# Residential Proximity, Duration, and Health-Related Quality of Life: Insights from the Fernald Cohort

**DOI:** 10.3390/ijerph22050738

**Published:** 2025-05-07

**Authors:** Sara Burcham, Wei-Wen Hsu, Sharon L. Larson, Jack Rubinstein, Susan M. Pinney

**Affiliations:** 1Department of Environmental and Public Health Sciences, Division of Epidemiology, University of Cincinnati College of Medicine, Cincinnati, OH 45267, USA; pinneysm@ucmail.uc.edu; 2Department of Environmental and Public Health Sciences, Division of Biostatistics and Bioinformatics, University of Cincinnati College of Medicine, Cincinnati, OH 45267, USA; hsuwe@ucmail.uc.edu; 3Jefferson College of Population Health, Thomas Jefferson University, Philadelphia, PA 19107, USA; sharon.larson@jefferson.edu; 4Department of Internal Medicine, Division of Cardiovascular Health and Disease, University of Cincinnati College of Medicine, Cincinnati, OH 45267, USA; rubinsjk@ucmail.uc.edu

**Keywords:** hazardous waste, well-being, health surveys, SF-36, community health, epidemiology

## Abstract

The impact of living near environmental contamination on the health-related quality of life (HRQoL) is not well understood. This study examined the impacts of the residential proximity (RP) and time spent near a former Department of Energy uranium processing facility (located in Fernald, Ohio) on the surrounding community’s HRQoL outcomes. A cross-sectional analysis was conducted using the data collected from participants using the Short Form-36 survey at the time of the enrollment in the Fernald Medical Monitoring Program (*n* = 7957). Mental and physical component summary scores (MCS and PCS, respectively) were computed for each participant. The scores were compared among the cohort participants, stratified by the RP to the facility and to the U.S. general population. Multivariable linear regression analyses were performed to identify associations between the RP from the facility, duration at residence, participant characteristics, and HRQoL. The adults and minors at enrollment (MAEs) living within two miles of the facility reported slightly lower MCS scores compared to those of residents who lived >2 miles from the facility, after controlling for confounding variables (adults: marginal effect (ME): −0.553, *p*-value: 0.002, MAEs: ME: −1.594, *p*-value: 0.040). The RP had a significant association with PCS scores among adults but not with the MAEs (adults: ME: −0.456, *p*-value: 0.010). No significant relationship was observed between the duration at residence and HRQoL. Considering the association between the RP and HRQoL in the Fernald cohort, integrating a health perception screening tool into community healthcare practices would benefit residents near environmentally contaminated sites to capture the variable nature of perceived health over time.

## 1. Introduction

Chronic environmental contamination (CEC) refers to the prolonged exposure to elevated levels of hazardous substances [1]. This contamination arises from the accumulation of physical, chemical, biological, or radiological pollutants, often stemming from human-made commercial or industrial activities [2]. The local community may experience direct exposure by consuming contaminated water or inhaling polluted air, leading to toxicological harm. Epidemiological studies have established a link between environmental pollutants and various adverse health outcomes in both adult and pediatric populations [3,4]. Apart from direct toxicological exposure, the presence of CEC may adversely impact residents’ mental and physical qualities of life by increasing neighborhood traffic; causing property damage, loss, or devaluation; generating light and noise residues; and triggering pathophysiological responses, such as worry, fear, stress, and anxiety, because of perceived risks to themselves or their loved ones [5]. Previous studies have established that residing near environmental contamination is linked to small-to-medium changes in psychological well-being, including anxiety, stress, depression, and post-traumatic stress disorder symptoms [1,6,7,8].

The impacts of environmental contamination extend globally, affecting both developed and developing countries undergoing urbanization, industrialization, and other human activities, such as the use of pesticides in agricultural operations. Intentional mitigation of pollutants is necessary to prevent health inequalities in morbidity and mortality within the affected populations [9]. Underdeveloped countries are particularly vulnerable because of factors including poverty, inadequate technological investments, and weak environmental legislation, contributing to elevated pollution levels [10].

In this study, the residential proximity (RP) is defined as the minimum distance between a resident’s closest primary residence and the contaminated site’s boundary. The RP serves as a proxy for perceived environmental risk and has been widely used in environmental epidemiology, economics, and health geography. Perceived environmental risk, in this study, refers to residents’ assessments of the likelihood of harm, loss, or negative impacts to themselves, their families, or their property from environmental hazards. Prior studies have employed the RP both as a measure of the perceived risk [11,12,13,14,15] and through the use of concentric rings around a site of interest [16,17,18,19,20,21]. Research has demonstrated that living near a CEC site influences threat perception [6,13,14,22,23,24] and negatively impacts mental well-being [7,25]. Individuals often respond to environmental hazards according to personal, subjective appraisals that can differ from objective exposure measures [26,27,28,29]. Environmental risk perception is shaped by limited and uncertain information, common in communities impacted by CEC, and is influenced by both hazard characteristics and personal beliefs [30]. The duration at the primary residence refers to the longest period spent at the identified residence, and some studies have shown that the length of time spent near CEC may negatively affect mental well-being because of perceived potential increases in toxicant exposure [31,32]. These objective exposure measures are relatively simple to ascertain and hold practical implications for public health prevention and remediation activities. For instance, these exposure definitions help to identify affected community boundaries and allocate resources when the full extent of the direct environmental exposure from chemical contamination is uncertain.

In the current literature, a gap exists regarding the impacts of CECs on the psychological well-being of children and adolescents who have grown up near contaminated environmental sites. Children represent a unique demographic when it comes to understanding the effects of toxicological events on physical, behavioral, mental, and emotional health. This may be the case because children often lack the learned experience for how to cope with chronic stress and may struggle to articulate their concerns because of limited vocabulary and coping skills [33]. Additionally, children’s attachment to their surroundings influences their senses of identity, security, and belonging [34]. Direct impacts on children from such sites may include a decrease in contamination-free green spaces, exposure to the illness or death of family members, forced relocation, or disruptions in schooling [33]. Furthermore, children may experience secondary effects because of their parents’ fears regarding the contaminated site.

Thus, the current study aims to investigate whether the RP to a former uranium processing facility (the Feed Materials Production Center (FMPC) and the duration of residence are significant risk factors for a reduced health-related quality of life (HRQoL). The RP and duration near the CEC site may be psychologically harmful because residents perceive the facility’s presence as being negative, a sign of a potential health threat or as a source of stress, not as a result of being disproportionately exposed to pollutants, as most residents living near the former uranium production facility in the village of Fernald, Ohio were not aware of their actual chemical exposures. The HRQoL serves as a metric for an individual’s subjective mental and physical well-being. Self-reported generic health measures, such as the HRQoL, are considered as the gold standard for assessing well-being in a population and are ideal for capturing the fluctuating nature of the psychological distress experienced by an individual over time [35]. Generic health measures are not intended to serve as substitutes for traditional measures of clinical endpoints but rather to assess functional status and well-being, appreciated universally and non-specific to age or disease or treatment status [36]. The second study aim is to assess the impacts of the RP and duration for children on the HRQoL outcomes of the same people as adults (≥18 years) and the associated risk factors for this sub-population.

This study utilizes data collected through the Fernald Community Cohort (FCC) [37]. The program’s objective was to conduct periodic screenings and health exams to alleviate residents’ concerns about cancer and to detect diagnoses early in the disease progression [38]. The cohort was monitored with clinical examinations for 18 years, and even after the conclusion of the medical monitoring in 2008, some members continued to participate with questionnaire data collection. The FCC serves as an ideal study population because of its substantial number of participants (*n* = 9782), their cooperation with the program, and the extended follow-up period.

Our primary objective is to investigate whether persons living near a CEC site constitute a vulnerable population to a reduced HRQoL. We hypothesize that individuals who lived in close proximity to the former uranium processing facility for a long duration of time are more likely to experience reduced HRQoL outcomes.

## 2. Materials and Methods

### 2.1. Participants

The members of the FCC consisted of all the persons who lived within five miles of the FMPC borders for a minimum of a continuous two-year period between 1 January 1952 and 18 December 1984 (*n* = 9782). Former employees of the facility were excluded from the cohort because they had their own workers’ medical monitoring program. The workers’ descriptions are described elsewhere [39,40]. Recruitment for the FCC began in 1990 using local media (television, radio, and newspapers), direct contact, word-of-mouth referrals, and letters to members of local schools and religious facilities [38].

For the present study, participants who provided work addresses only (*n* = 353) or incomplete residential addresses (*n* = 107) were excluded from the analysis. To be included in the current study, the adult participants must have enrolled between 1990 and 1995 and included 7957 participants. The minor-at-enrollment (MAE) participants were included if they enrolled in the cohort as a child (<18 years) and matured into adulthood (18 years or older) so that they were eligible to complete the baseline questionnaire between the years 1991 and 1997 (*n* = 268).

### 2.2. Data Collection

Questionnaires were administered to participants that included demographics, medical history, and the Health Risk Appraisal (HRA) that consisted of questions related to self-reported feelings of life satisfaction and overall physical health. Residents were asked to report all the previous addresses where they had lived for at least three consecutive months within a five-mile radius from the FMPC borders since 1952 [41]. For each address listed, the residents were asked to list the start and end dates of their residency, as well as the years spent at each residence provided. The participants were asked to verify their residential addresses and associated dates at each residence at the time of their second questionnaire. A random sample of addresses was verified using records from the Hamilton and Butler County Recorder of Deeds offices. All the addresses were geocoded and categorized by concentric mile rings (0–5-mile radii) as a measure of the RP. The mile ring of the closest residential address was used as the value for the RP variable.

Physical examinations were administered to the medical monitoring participants every 2–3 years, regardless of their exposure or insurance status [42]. For the present study, data from physical examinations included the review of systems, an inventory of somatic complaints (count) obtained through a series of questions asked by the physician, weight (pounds), height (meters), and current medications prescribed for anxiety, depression, or psychosis (yes/no).

### 2.3. Health-Related Quality of Life Tools and Scoring

The health-related quality of life (HRQoL) is a multidimensional personal assessment of perceived mental and physical well-being and was measured in the FCC with the Short Form-36 (SF-36). This tool has been validated to assess generic, as opposed to disease-specific, self-rated health in the general U.S. population and is useful in comparing populations [43,44]. Participants who were children (<18 years) completed the SF-36 when they became 18 years old.

The SF-36 was developed as a result of the Medical Outcomes Survey (MOS), a four-year observational study designed to understand how components of the healthcare system affect patient outcomes across different healthcare settings [45]. Generic measures of well-being that include a broad spectrum of health concepts appropriate for groups differing in “diseases, severity, and comorbidity” have been in use as early as 1970 [35]. Since then, multiple versions of the HRQoL generic tool have been created, including the SF-12, SF-8, and the Veterans RAND-36 (VR-36) [46].

The SF-36 was selected for the FCC because of its simplicity in administration and scoring, as well as its ability to provide eight health domain scores and two principal component scores. Among the forty health concepts initially identified in the MOS, the SF-36 assesses eight: physical functioning, role limitations because of physical health, bodily pain, general health, vitality, mental health, social functioning, and role limitations because of emotional problems.

The mental component summary (MCS) and physical component summary (PCS) scores are aggregates of the eight health scores for the SF-36 and are scored on a scale of 0–100 [46,47]. Of the 36 questions asked in the survey, all but one are used to calculate the domain scores. The calculation processes for the MCS and PCS have been described elsewhere [43,48]. The domains formed in the SF-36 are scored on the 0–100 scale. For all the scales, lower values represent a lower health status, and higher values represent a positive effect.

### 2.4. Determinants of the HRQoL

The demographic covariates included sex (male/female), age at the time of the questionnaire (years), marital status (married/single/separated/divorced/widowed), annual household income (USD), and the highest education level attained (some high school/high school graduate/some college/college graduate/vocational training/professional degree). As the MAEs partook in a separate “kid demographic questionnaire” at the time of their enrollment to the FCC as children (<18 years of age), the variables marital status, income, and education level were not included in the analysis.

The behavioral covariates included “cumulative cigarette smoking pack years” (years, adults), “ever smoked cigarettes” (yes/no, MAEs), the frequency of alcohol consumption per week (the count of average drinks per week), the level of agreement with job satisfaction (strongly disagree, disagree, somewhat agree, or strongly agree), the average number of hours spent in physical activity each week (0–1, 1–2, or ≥3 h), the average number of hours of sleep per night (≤6, 7, 8, or ≥9 h), “ever lived with a Fernald employee” (yes/no), awareness of environmental hazards (none, rarely, some of the time, not at all, most of the time, or all of the time), and the perceived strength of social ties to friends and family members (weaker than average, not sure, about average, or very strong).

The clinical covariates included in the models were the body mass index (BMI) z-score; current use of medications intended to treat anxiety, depression, or psychosis, grouped into a “Yes” or “No” variable; current diagnosis of a chronic disease (cancer, heart disease, chronic bronchitis, or diabetes); self-reported knowledge of familial cancer (yes, no, or not sure); and the count of somatic complaints reported at the time of the physical examination.

For all the covariates with missing data, we implemented a multiple imputation of missing data with ten imputed datasets to reduce the bias [49]. We employed multiple imputation by fully conditional specification (FCS), as the FCS method is a valid approach for handling both categorical and continuous variables with missing data [50].

### 2.5. Statistical Analysis

Statistical analyses were conducted with SAS v. 9.4 [51]. Descriptive statistics for demographic characteristics were calculated for the adult and child participants at the time of the initial questionnaire and are displayed in Table 1. Range checks were performed on all the variables to identify unusual or extreme data points, followed by verification by checking the original questionnaire for values that fell in the 99th percentile of the distribution or were outside of the “expected range” of values to determine if they were valid or should be edited in the dataset [52]. The HRQoL mean and standard deviations were calculated, and the descriptive statistics for the cohort can be found in Figure 1 for all the participants who completed an SF-36 questionnaire. All the variables were evaluated for multicollinearity using Pearson and Spearman’s rank correlation coefficients, as well as polychoric correlations where ordinal variables are assessed and assumed to originate from an underlying continuous distribution [53]. No variables were highly correlated (≥0.40).

The HRQoL scores of the adult and MAE populations were compared to that of the U.S. general population. The U.S. general population data were collected between 1989 and 1990 by the National Opinion Research Center (*n* = 2472) at the University of Chicago and published as a part of the National Survey of Functional Health Status (NSFHS) [54]. The NSFHS obtained national normative data, using the SF-36 from the non-institutionalized general population. The sampled population consisted of individuals aged 18 and older, drawn from 2909 households. Although that study aimed for a balanced distribution of sexes, detailed racial and ethnic compositions were not specified in the final 1991 report. However, it is presumed that the sample demographics closely reflected those of the U.S. general population during that period.

As the MCS and PCS scores of the FCC’s adult population were identified to have a left-skewed distribution, a non-parametric test was performed to compare the HRQoL median scores among the participants, stratified by dichotomized mile group using the Kruskal–Wallis non-parametric test (Figure A1), descriptive statistics (Table 2), and pairwise comparisons using the Wilcoxon rank sum test (Table 3) [55,56]. The MAE sample sizes in the individual mile rings were too small to compare to the U.S. general population. The median HRQoL scores for ages 18–24 in the general U.S. population were used (*n* = 173) and compared to the FCC’s MAE scores (*n* = 268) [48].

Four separate multivariable linear regression models were run to evaluate the relationships between the RP and MCS and PCS for the adult and MAE cohorts. The generalized linear model (GLM), developed by Nelder and Wedderburn in 1972, was chosen to evaluate the systematic and random components of the statistical models after the MCS- and PCS-dependent model variables underwent a power transformation using a cubic transformation (e.g., MCS^3^) to meet the assumptions for the normality and homoscedasticity of the residuals [57]. The final models were chosen based on the forced entry approach with, first, the closest RP, sex, and age included in the “base model”, followed by the significant variables in a stepwise model selection approach [58].

We conducted multiple sensitivity analyses to determine the characterization of the independent variable with the most significant effect on the dependent variables. First, we investigated the impact of the duration at the closest residence as the primary explanatory variable. Subsequently, we assessed the effect of the total duration within the exposure domain, accounting for participants who relocated within the five-mile radius between the facility’s construction in 1950 and the administration of the SF-36 questionnaire. We evaluated a multiplicative variable, the closest residence, reverse scored, multiplied by the duration. Finally, we tested the dichotomization of the independent variable into participants living ≤2 miles and those living ≥3 miles from the facility’s borders.

As the dependent HRQoL variables were cubed in the regression analysis, to present the results on the original SF-36 scale (0–100), we performed an estimated marginal effect (ME) calculation [59,60]. The ME statistic is interpreted as the effect of the estimated difference between living in the closest-mile ring and the furthest-mile ring on the dependent variable, with all the other variables held constant. The ME of x_1_ on Y was estimated using the full range of predicted Y values from the generalized linear models’ output that considers all the model covariates.

## 3. Results

### 3.1. Baseline Characteristics

Of a total of 8788 adult FCC participants, 831 participants were missing a complete residential address or received their first examination past the designated enrollment period of 1990–1995, with 7957, in total, adult participants remaining. There were 84 participants who had at least one of the SF-36 domains missing, representing missing or incomplete questions, and an SF-36 MCS/PCS score could not be generated (*n* = 7873). Of a total of 1006 child and adolescent participants who were enrolled in the FCC, 26 were missing a complete residential address, with 980, in total, child participants who completed the baseline examination as a child. Only the MAEs who completed the SF-36 were included in the present study, leaving 268 MAEs for the initial questionnaire analysis.

### 3.2. Initial Questionnaire Characteristics

Of the 7957 adult participants, 84 were missing data for at least one of the SF-36 health domains. Their scores were excluded when calculating the principal component summary scores. As a result, 7873 participants were included in the regression analysis.

The average age of the adult participants who completed the initial questionnaire was 44.5 ± 14.9 years, which is older than the median age of U.S. citizens reported in the 1989 census (32.1 years) [61]. The FCC’s adult population was 56.2% female—slightly higher than the U.S. population’s average of 51.3%, though the sex distribution varies by state. Additionally, the FCC sample was 99.6% White, reflecting the demographics of the surrounding five-mile exposure area. In contrast, the general U.S. population at the time was 12.2% Black and 7.9% Hispanic; groups not represented in the FCC sample.

The participants living in the closest-mile ring had resided there for an average of 13.5 ± 9.4 years. Overall, 72.5% of the participants were married. A total of 2138 participants (28.7%) reported an annual household income of between $20,000 and $34,999. Sleep habits showed that 3396 participants (43.2%) averaged 7 h. of sleep per night. For education, 2965 participants (38.0%) reported that a high school diploma was their highest level of education. The participants reported consuming an average of 2.9 alcoholic drinks per week and had a mean of 8.4 smoking pack years at the time of the initial questionnaire. At enrollment, 13.7% of the adult participants reported having a chronic illness, including heart problems, cancer, chronic bronchitis, or diabetes.

From the 980 child participants who participated in the baseline physical examination, 731 (74.6%) returned as adults (referred to as the “MAEs”), although only those participants who completed the SF-36 are included in the present analysis (*n* = 268). As the demographic questionnaire was not administered to the MAEs, these data are not available for summary statistics. All the available characteristic summary data for the FCC’s participants can be found in Table A1 and HRQoL scores for the adults stratified by self-reported chronic illness at the time of enrollment can be found in Table A2. 

In the evaluation of the adults’ HRQoL median scores by demographic, on average, male participants reported higher MCS and PCS scores compared to those of females (MCS: 52.8 vs. 51.3, PCS: 50.4 vs. 49.5). The younger adult participants reported lower MCS values than older adults, whereas the younger adults reported higher PCS values when compared to those of the older adult participants, on average, in the cohort. In general, the participants who reported being separated, widowed, or divorced reported lower MCS and PCS values compared to those of the participants who were single or married, while the participants with higher annual household incomes and those with higher educational achievements reported better MCS and PCS scores. The participants who self-reported having a chronic illness had lower MCS and PCS scores compared to those of the participants who did not report a chronic illness (Table 2). Although the group differences in Table 2 are statistically significant, they are modest in magnitude.

The unadjusted descriptive summary scores of the eight health domains and principal component summary scores identified that among the adult population, the FCC had lower HRQoL scores than the general U.S. population in all the health domains except for social functioning (t(7950) = 15.36, *p* ≤ 0.001). Interestingly, the MAEs had lower health domain scores for bodily pain, general health, mental health, and vitality, compared to those of the general U.S. population, ages 18–24 (Figure 1).

### 3.3. HRQoL Differences Within the Cohort and Comparisons with the U.S. General Population

The results from the two-sample Wilcoxon rank sum test found small but statistically significant differences between HRQoL scores when the RP was dichotomized into <2/>2 miles from the FMPC in the adult cohort, but no significant difference was observed in the MAEs’ HRQoL scores. Figure A1 displays these comparisons using a non-parametric analysis-of-variance test for unadjusted HRQoL scores by the RP in the cohort.

The HRQoL scores for the adult and MAE populations compared to those for the general U.S. population suggest there were significantly different MCS and PCS scores among the adult population, but only the PCS scores were statistically significant for the MAEs compared to those of the general U.S. population for young adults, as displayed in Table 3. Among the FCC’s adults, those living in Mile Ring 1 had the greatest difference in the median MCS scores compared to those of the general U.S. population [median difference = −3.49, 95% CI: [−4.32, −2.62]. The adult participants living in Mile Ring 2 had the greatest difference in the median PCS scores compared to those of the general U.S. population [median difference = −5.07, 95% CI: −5.54, −4.64]. The full-sample MAE’s median PCS score was significantly lower than the general U.S. population’s PCS score [median difference = −1.83, 95% CI: −2.63, −1.03].

### 3.4. The Closest Residential Proximity to the Environmental Contamination and the Corresponding HRQoL Outcomes

In the base regression models (i.e., adjusted for age and sex only), for the FCC’s adult participants, the closest RP had significant negative associations with the MCS (*p*-value 0.004, R^2^ = 0.038) and PCS (*p*-value 0.005, R^2^ = 0.119) scores. No significant association was observed for the MAE participants.

In the fully adjusted regression models for the FCC’s adult and MAE participants, the closest RP (0–1 miles) exhibited negative associations with the MCS and PCS scores when the reference group was those residents living the furthest away from the facility (4–5 miles), although these results were not statistically significant (Table A3).

### 3.5. The Results of the Sensitivity Analysis

The sensitivity analysis revealed the model that produced the largest effect size estimate for both the MCS and PCS outcomes among the adult participants was the dichotomized mile ring, after controlling for the model’s covariates. There was a risk of a 0.527 unit decrease in MCS scores for the participants who lived <2 miles from the FMPC compared to those participants who lived >2 miles from the facility, after controlling for the model’s covariates, age, sex, ever lived with a Fernald employee, job satisfaction, strength of social ties, average hours of sleep per night, recent misfortune, currently prescribed a medication for mental health, count of somatic complaints at the exam, cigarette smoking pack years, and witness to violent events in the past year, and the interaction term age*sex (*p*-value: 0.001). Among the FCC’s adults, there was a risk of a 0.418 unit decrease in PCS scores for the participants who lived <2 miles from the FMPC, after controlling for the model’s covariates, age, sex, annual household income at the baseline, awareness of environmental hazards, the highest education level achieved, self-reporting of a chronic illness, count of somatic complaints at the exam, the frequency of physical activity, and BMIz.

Among the MAEs, when the RP was dichotomized, it was significant in relationship to the MCS (*p*-value: 0.040), after controlling for the model’s covariates, age, sex, job satisfaction, strength of social ties, currently prescribed a medication for mental health, ever smoked cigarettes, and witness to violent events in the past year. There were no statistically significant associations between the RP or duration and the PCS among the MAEs, as displayed in Table 4. These results indicate a modest yet statistically significant effect of living in close proximity to the facility on the HRQoL at the time of the first SF-36 questionnaire in a fully adjusted model for adults and MAEs.

### 3.6. Models’ Covariates

In the models evaluating the relationships between the residential exposure to the CEC site and the MCS in both adults and MAEs, several significant findings emerged. Females were more prone than males and those prescribed a mental health medication were more likely to have lower MCS scores (*p*-value < 0.001). Experiencing two or more losses in the previous year was associated with reduced MCS scores compared to those where no such losses were experienced (*p*-value < 0.001). Additionally, cumulative cigarette smoking pack years (for adults) and ever smoking (for MAEs) were linked to diminished MCS scores. Disagreement and strong disagreement with job satisfaction, or unemployment, showed negative associations with MCS, as did exposure to violence in the previous year compared to no exposure. A self-reported family history of cancer, a self-reported chronic disorder, and the average alcohol consumption per week were not significantly associated with the MCS for adults or MAEs. The parameter estimates predicting cubed MCS and PCS values are described in Table A3.

In the models predicting the PCS, cigarette smoking, the body mass index z-score, self-reported physical activity levels, and the count of somatic complaints at the closest physical examination were significant among the adults but not for the MAEs. Age, sex, and a weak or an unsure strength of social ties were significantly associated with reduced PCS values among the MAEs.

In the models evaluating the relationships between the residential exposure (the closest residence, dichotomized closest residence, cumulative duration, and reverse scored proximity × duration) and the PCS and MCS values, the following covariates remained significant: age, sex, job satisfaction, strength of social ties, current medication to treat a mental health disorder, ever smoked cigarettes, and witness to a violent event in the previous year.

An interaction term, age*sex, had significant associations with MCS and PCS outcomes in the adult cohort. In predicting the MCS, an increase in age of one year in females resulted in higher MCS scores (*p*-value 0.045). In predicting the PCS, an increase in age, if the participant was male, resulted in higher PCS scores (*p*-value 0.005).

## 4. Discussion

In this study, we found that living in close proximity to the former uranium processing facility was significantly associated with lower self-reported mental and physical HRQoL scores at the baseline examination for both adult and MAE participants. This association was identified while comparing participants living within two miles of the facility to those living three or more miles away, with the models adjusted for confounding variables. Notable differences in mental and physical component summary scores were observed, with scores generally lower than those of the general U.S. population, except for the MCS scores among the MAEs. Sensitivity analysis was performed to optimize the characterization of the primary independent variable by examining both the RP and time spent within 5 miles of the facility to explore their effects on the HRQoL.

Our initial hypothesis predicted that residents living the furthest from the facility would report the best health. Surprisingly, the FCC’s residents within the 2–3-mile ring reported the highest mental and physical self-rated health. Historical research revealed that a large landfill and two major highways outside the 4–5-mile ring might have negatively impacted the community’s self-rated health. Conversely, residents within the 2–3-mile ring enjoyed proximity to Miami Whitewater Forest Park and a golf course, which may have positively influenced their self-rated health. Previous studies have suggested that landfills and highways may increase noise levels, diminish environmental aesthetics, and heighten pollution perception, thus potentially reducing the self-rated health of the FCC participants residing in the furthest-mile ring from the site [62,63]. Access to greenspace and a community park may have conferred a protective effect on residents’ self-rated health within mile ring 2–3 [64]. We reclassified the primary independent variable into two groups: residents within 2 miles of the facility and those 2–5 miles away, as a part of a sensitivity analysis to test the robustness of our findings. This categorization was significantly associated with HRQoL outcomes.

For the MAEs, the RP also significantly impacted mental well-being. MAEs living within 2 miles of the facility experienced a 1.594 decrease in the MCS score compared to those living 2–5 miles away (MCS ME = −1.594, *p* < 0.001), on average, after adjusting for other covariates. However, no significant effect on PCS scores was observed for MAEs. The greater impact of the RP on the MCS in MAEs compared to adults may be explained by differences in susceptibility to chemical and non-chemical stressors. According to the critical period hypothesis, exposure to environmental chemicals during key developmental stages can lead to irreversible health effects, such as neurotoxicity and behavior disorders, because of the brain’s heightened plasticity during these critical windows [65]. Regarding non-chemical stressors, children and adolescents may have been particularly vulnerable because they lacked the emotional coping mechanisms to process environmental stressors, articulate their concerns, or regulate their emotions. Additionally, they may have experienced cumulative stress because of their household members’ distress and disease burden, such as witnessing illness, forced relocation, or the loss of a loved one, which they may have attributed to the presence of the CEC site in their community [66]. Further, studies indicate that children differ from adults in their access to and understanding of media coverage, such as television and newspapers, related to environmental events, as well as their comprehension of the severity of such stressors [67,68]. Previous studies have also shown that the opposite effect, exposure to residential greenspaces, has an improved effect on child and adolescent mental well-being [69]. These results highlight the differential impacts of the RP on mental and physical health across age groups in relation to the time of the FCC operations and cohort time frame.

Our study examined the duration of residence in multiple ways as a part of the sensitivity analysis and found it notable that none of the approaches used to characterize the duration was statistically significant in the HRQoL models for either adults or MAEs. We initially hypothesized that the duration at the closest residence and the total duration in the exposure domain would reflect a “sense of place” (SoP)—a broad psychological concept that includes the concepts of “place attachment” and “place identity” theories that may be protective in the final models [70,71]. Additionally, other theories, such as “rooting” or “spatial anchoring”, which are reinforced by residential stability, might have also influenced HRQoL scores [72]. This framework suggests that individuals’ perceptions of their environment are subjective and influenced by the time spent at a particular location. Over time, memory formation and social connections, such as relationships with Fernald workers and neighbors near the site, may have shaped the participants’ experiences and perceptions of their surroundings. However, previous studies have identified mixed results, such as those described by Prior and colleagues in 2019, who found that proximity was significantly associated with the residents’ worry about a contaminated site whereas the SoP was not [12]. A recent study found that ecological grief diminished as FCC residents lived further from the site [73]. In the predominantly farming community, the residents likely felt the pronounced presence of ecological loss. Additionally, a case-control study identified minimal differences in PTSD symptoms among children living near the Fernald FMPC over time compared to those of a control group, though parental psychological functioning significantly contributed to PTSD symptoms, a factor not controlled in this study [68].

### 4.1. Comparisons with Other Studies

Our study differs from other environmental epidemiology studies in that we included children who lived near the site and followed them into adulthood, capturing their self-perceived health through baseline SF-36 questionnaires. Recent research has begun to address climate anxiety, environmental injustice, and stressors affecting children’s psychological well-being, identified as priority research areas in a 2021 National Academies of Sciences workshop [74]. Sensitivity analyses helped to identify the best method for characterizing the independent variable in our models and revealed secondary risk factors significantly affecting self-rated health, including sociodemographic, social, and clinical conditions.

Among the secondary risk factors, the perceived strength of social ties and job satisfaction consistently correlated with the HRQoL. Variables such as the BMI, somatic complaints, and annual household income showed strong associations with physical health. These findings align with prior research on subjective well-being in populations affected by environmental contamination [75,76]. A meta-analysis by Faragher and colleagues (2005) found a high degree of correlation between job satisfaction and psychological problems, including burnout, anxiety, and depression, proposing that long and inflexible work hours, demanding timelines, feelings of a lack of control over workloads, and job insecurity may mediate the relationship [77]. A systematic review of the literature examining the relationship between income and self-rated health found that the variable yearly household income was a good predictor of self-rated health in longitudinal study designs [78]. Social capital is known to enhance community coping, resilience, and action, while job satisfaction has been linked to psychological problems, such as burnout and anxiety. Income influences the quality of life, with higher income generally being associated with better self-rated health because of increased material welfare and healthcare options.

Some studies have explored the role of actual chemical exposure in HRQoL outcomes [66,79]. Residents in affected areas may have faced contamination via air emissions, groundwater, and soil [80]. However, our study did not include actual exposure effects, as residents were unaware of their chemical exposure. We focused on perceived exposure’s potential impact on psychological well-being, mediated through other factors, such as a cancer diagnosis, which were adjusted for in our models.

Race/ethnicity was not included in our multivariable models because of the predominantly White population of the Fernald cohort, mirroring Hamilton and Butler Counties’ demographics [81]. Despite this homogeneity, the key finding, that environmental contamination increases diminished mental and physical well-being, is relevant across all communities, including racial or ethnic minorities and socioeconomically disadvantaged groups. This message is especially pertinent to racial or ethnic minorities, linguistically isolated individuals, and those with less than a high school education, as they are disproportionately affected by the presence of superfund sites, which are remedial sites listed on the Environmental Protection Agency’s (EPA’s) National Priority List [82]. We recommend that future research prioritize examining the impact of the RP to a CEC site on more racially and ethnically diverse populations to better understand potential disparities in health outcomes.

Consistent with previous research, we observed variations in SF-36 health domain and summary scores between the participants with and without a chronic illness, as displayed in Table A2. A 2009 review found significant differences in PCS scores because of co-morbid conditions, like heart failure and diabetes, but smaller differences in MCS scores [83]. Another study found that adult cancer survivors (*n* = 392) had significantly lower SF-36 scores compared to that of the U.S. general population [84].

### 4.2. Strengths

This study has several strengths: a large sample size, a rigorous data collection protocol, and the use of the validated SF-36 tool for HRQoL measurements. Although the medical community often relies on categorical diagnoses of specific mental disorders, such as those outlined in the Diagnostic and Statistical Manual of Mental Disorders (e.g., DSM-5) or other taxonomies, like the International Classification of Diseases (e.g., ICD-10), the SF-36 tool employed in this study offers an alternative research framework. It emphasizes dimensional assessments of psychological well-being, acknowledging that symptoms exist along a spectrum that fluctuates over time. The FCC cohort’s large sample population (n = 9782) and high response rate enhanced the statistical power. We also explored various covariates affecting the HRQoL, including mental health medications, a research area that is underexplored. Finally, the FCC investigators have an active program of sharing data and biospecimens, which are available to other investigators upon request to expand on the present study’s findings and conduct future research on the health impacts of living near CECs. Information about research opportunities and accessing data and biospecimens can be found at https://med.uc.edu/depart/eh/research/projects/fcc/research-opportunities (accessed on 30 March 2023) and https://med.uc.edu/depart/eh/research/projects/fcc/sharing-of-fcc-data-and-biospecimens (accessed on 30 March 2023).

### 4.3. Limitations

Our study’s cross-sectional design limits the analysis to baseline data. To address this limitation, a future retrospective longitudinal study will assess HRQoL changes over time in the FCC’s adult population. This study will incorporate mixed-effect regression and survival analysis techniques to appropriately account for repeated measurements per participant.

The racially homogeneous study population may limit generalizability, but this allowed for an evaluation of the RP to environmental contamination without racial or poverty-related confounding effects. The RP and duration at each address were collected through self-reported questions, which may have been subject to recall bias. To mitigate this, FCC investigators employed multiple strategies, including providing memory cues, such as maps of the Fernald region, notable landmarks, and follow-up questions, to help participants to accurately recall their addresses and time periods of residency.

We chose not to evaluate the relationship between the subjective risk perception and HRQoL outcomes, as the study population was not blinded to the exposure and, thus, may have contributed to reporting bias. Instead, we used surrogate variables, such as the RP and duration of the residence, to generate objective measures of exposure to the FMPC. This approach aligns with common methods of characterization seen in recently published systematic and narrative reviews [6,7].

Direct environmental contaminant exposures were not included in the analysis. However, this study aimed to examine the relationship between the perceived risk of exposure, using proximity as a surrogate, and the HRQoL, rather than the effects of direct contaminant exposure. This approach was necessary, as residents were unaware of their actual pollutant exposure during this time period. A mile ring would not have been an appropriate surrogate variable to use for the actual pollutant exposure, as previous dosimetry assessments have shown that estimated uranium and radon concentrations were not dispersed equally around the facility but, rather, were determined by other exposure-related factors, such as the calendar year of the exposure, the location relative to the prevailing wind direction, and the use of private wells or cisterns for drinking water [80]. It is plausible that personal knowledge of these factors could have influenced the perceived risk among the participants; however, these factors were not included in the present study.

## 5. Conclusions

The findings of this study indicate that residing close to a CEC site may diminish the HRQoL. Despite the residents’ experiencing a perceived risk of harm from the former uranium processing facility, the duration variable was not a significant risk factor for reduced HRQoL among the FCC’s adult or MAE participants. This finding suggests that attachment to place and residential stability might offer protective effects on the HRQoL. Specifically, our results suggest that factors such as a landfill and busy local highways and interstates, which are highly visible to the residential community, may have negatively impacted residents living 4–5 miles from the facility. Conversely, residents living 2–3 miles from the facility had more exposure to greenspace, which potentially contributed to better self-rated health.

Chronic environmental contamination is a widespread global public health issue. Therefore, addressing it requires a collaborative effort involving community members, manufacturing and industry stakeholders, policymakers, and medical practitioners. By working together, we can mitigate the risk of diminished HRQoL in affected communities.

## Figures and Tables

**Figure 1 ijerph-22-00738-f001:**
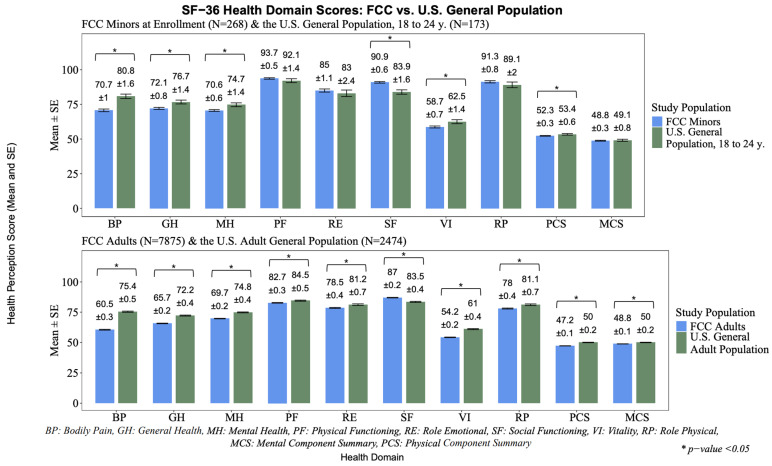
HRQoL domain and summary scores among the FCC participants and the U.S. general population, with *t*-test comparisons between the groups.

**Table 1 ijerph-22-00738-t001:** Demographic characteristics of the study participants by age group at the time of the initial questionnaire.

	Adults		MAEs	
Characteristic	*n*	Value	*n*	Value
Age (years, mean ± SD) ^†^	7957	44.5 ± 14.9	268	19.5 ± 1.3
Female sex (%) ^†^	4470	56.2	138	51.5
White race (%) ^†^	7926	99.6	265	98.9%

Notes: *n*, sample size; SD, standard deviation; MAEs, minors at enrollment; ^†^ no missing data for age, sex, or race.

**Table 2 ijerph-22-00738-t002:** Descriptive HRQoL summary scores of FCC’s adult participants by demographic characteristic at the time of the enrollment.

Characteristic	*n*	MCS		PCS	
		Mean/Median	SD/IQR	Mean/Median	SD/IQR
Sex *					
Female	4431	48.1/51.3	10.9/14.1	46.9/49.5	10.9/13.9
Male	3442	50.0/52.8	10.1/11.8	47.7/50.4	10.1/12.2
Age *					
<25	725	48.5/50.9	9.6/10.9	51.7/53.4	7.6/8.1
25–43	3349	47.6/51.1	10.7/13.7	49.6/51.7	8.8/10.5
44–55	1902	49.2/52.5	10.7/12.8	47.1/49.6	10.2/13.0
>55	1897	51.0/54.1	10.4/13.1	41.6/43.8	11.7/18.3
Marital status *					
Single	1076	48.6/51.3	9.9/12.1	50.6/52.7	8.4/9.2
Married	5654	49.5/52.5	10.3/12.2	47.2/49.9	10.2/12.9
Divorced Separated Widowed	1062	45.9/49.2	12.4/18.1	44.4/47.0	11.9/17.6
Household income *					
<$20,000	1955	46.4/49.2	12.2/18.0	43.1/45.6	12.2/19.0
$20,000–34,999	2137	48.9/51.9	10.4/13.6	47.5/49.6	9.8/12.5
$35,000–49,999	1708	49.5/52.1	9.8/11.2	49.3/51.5	9.0/10.5
$50,000–74,999	1182	51.2/53.6	8.7/8.3	49.8/51.6	8.1/9.7
>$75,000	474	51.2/54.0	8.7/9.4	50.7/52.6	7.9/9.6
Educational attainment *					
Some HS	1223	46.9/49.8	12.0/17.6	41.7/43.8	11.9/19.4
HS graduate	2967	49.0/52.1	10.5/13.0	47.3/49.6	10.0/12.4
Some college	2099	48.8/51.8	10.5/13.3	48.4/50.7	9.7/12.1
College graduate	1047	50.2/53.2	9.5/9.8	50.5/52.5	8.7/9.5
Postgraduate	460	51.7/53.9	8.8/8.3	50.3/52.6	8.5/10.2
Self-reported chronic illness *					
Yes	1067	46.8/49.5	12.0/18.1	39.0/39.8	11.7/18.9
No	6806	49.2/52.3	10.3/12.4	48.5/50.9	9.6/11.5

Notes: *n*, sample size; MCS, mental component summary; PCS, physical component summary; SD, standard deviation; IQR, interquartile range; HS, high school; * all the comparisons were statistically different at a *p*-value of ≤0.001 (comparisons between the groups with the overall Wilcoxon rank test).

**Table 3 ijerph-22-00738-t003:** Descriptive HRQoL summary scores by residential proximity and the one-sample Wilcoxon signed rank test of the FCC’s participants (at the time of the first SF-36 questionnaire) compared to those of the U.S. general population (1989–1990).

FCC Population	SF-36 Component Summary Domain	Mile(s) from the FMPC	Median	Q1–Q3	*p*-Value	General U.S. Population Median †
Adults	MCS	1	49.03	48.2–49.9	<0.001	52.52
		2	49.13	48.6–49.6	<0.001	
		3	51.47	50.9–51.9	<0.001	
		4	51.18	50.6–51.7	<0.001	
		5	50.80	50.2–51.3	<0.001	
		total	50.25	49.9–50.5	<0.001	
	PCS	1	47.95	47.3–48.6	<0.001	52.64
		2	47.57	47.1–48.0	<0.001	
		3	48.79	48.2–49.4	<0.001	
		4	49.72	49.1–50.3	<0.001	
		5	48.71	48.1–49.3	<0.001	
		total	48.42	48.2–48.7	<0.001	
MAEs	MCS	total	50.2	49.1–51.2	0.074	51.13
	PCS	total	53.2	52.4–54.0	<0.001	55.03

Notes: FCC, Fernald Community Cohort; SF-36, Short Form-36; FMPC, Feed Materials Production Center; Q1–Q3, interquartile range; MCS, mental component summary; PCS, physical component summary; † median HRQoL scores for ages 18–24 in the general U.S. population were used in the comparison to the FCC’s MAE scores.

**Table 4 ijerph-22-00738-t004:** Primary predictor’s estimated marginal effect on HRQoL outcomes in the fully adjusted models with the models’ goodness-of-fit values.

Description	Adults (*n* = 7873)		MAEs (*n* = 268)	
	ME	R^2^	ME	R^2^
Model 1: MCS ß Closest Proximity (ref: 4–5 miles)	−0.388	0.294	−1.624	0.316
Model 2: MCS ß Cumulative Duration	0.004	0.292	0.102	0.301
Model 3: MCS ß Closest Proximity × Duration	0.005	0.293	0.007	0.297
Model 4: MCS ß Dichotomized Proximity (ref: >2 miles)	−0.553 ***	0.293	−1.594 **	0.308
Model 5: PCS ß Closest Proximity (ref: 4–5 miles)	−0.280	0.354	−2.013	0.123
Model 6: PCS ß Cumulative Duration	−0.008	0.353	0.102	0.120
Model 7: PCS ß Closest Proximity × Duration	−0.001	0.353	0.010	0.115
Model 8: PCS ß Dichotomized Proximity (ref: >2 miles)	−0.456 **	0.354	−0.157	0.114

Notes: ** *p*-value < 0.05; *** *p*-value < 0.001; n, sample size; ME, marginal effect; R^2^, coefficient of determination; MCS, mental component summary; PCS, physical component summary; ref, reference group. Adults’ linear regression model covariates: MCS ß age, sex, ever lived with a Fernald employee, job satisfaction, strength of social ties, average hours of sleep per night, recent misfortune, currently prescribed a medication for mental health, count of somatic complaints at the exam, cigarette smoking pack years, and witness to violent events in the past year, and the interaction term age*sex. PCS ß age, sex, annual household income at the baseline, awareness of environmental hazards, the highest education level achieved, self-report of a chronic illness, count of somatic complaints at the exam, the frequency of physical activity, and BMIz. MAE’s linear regression model covariates: MCS ß age, sex, job satisfaction, strength of social ties, currently prescribed a medication for mental health, ever smoked cigarettes, and witness to violent events in the past year. PCS ß age, sex, recent misfortune, and strength of social ties.

## Data Availability

Data sharing must be approved by the Fernald Community Alliance upon request.

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
