# Peer review of "Residential Proximity, Duration, and Health-Related Quality of Life: Insights from the Fernald Cohort"

_ijerph, 2025, doi:10.3390/ijerph22050738_

Round 1
Reviewer 1 Report
Comments and Suggestions for Authors
This manuscript reports the results of a study comparing residential proximity to a contaminated property with scores on a longitudinal mental and physical health survey. The authors use multiple regression to examine this relationship, arguing that people residing closer to the contamination exhibit diminished mental and physical well-being compared to those living farther away. This is an interesting study that makes use of an existing dataset and shows how previously published data can be re-analyzed for new insights into how people may experience chronically contaminated environments. I think this is an important study, even if simply for the research design. At the same time, the manuscript has challenges that exceed what one might consider to be a “major revision,” and so I cannot recommend it for publication. I encourage the authors to rework the manuscript and seek publication in the future.
Overall: What is the sample size for this study? The abstract (line 20) says n=7875, the methods (line 131) say it’s n=7957, and the results (line 260) say it’s n=7975. This is very confusing.
Introduction: “Pollutants” and “contaminants” are distinct terms with different meanings (e.g., all pollutants are contaminants, but not all contaminants are pollutants), but they are used interchangeably throughout the introduction.
Lines 37-39: “This contamination arises from the accumulation of non-indigenous physical, chemical, or radiological pollutants...” should be updated. “Non-indigenous” only captures some instances of CEC; for example, in some kart environments, arsenic contamination is “chronic.” Also, “biological” is missing from the list of pollutants; there are many documented instances of chronic contamination in urban environments (e.g., stormwater ponds) of E. coli and other coliform bacteria. Perhaps the sentence can be re-worded to: “This contamination arises from the accumulation of physical, chemical, biological, or radiological pollutants...”
Lines 43-50: The literature on “chronic environmental contamination” could be updated. A number of key studies in the social sciences were published in 2023/2024.
Lines 57-58: “Through media reports, the general public is aware of these health effects of environmental contamination.” There is a substantial literature that does not support this claim.
Lines 61-64: the authors argue that proximity (distance) to a contaminated site is a reliable proxy for people’s perception of risk (regardless as to whether or not they know about the existence of contamination or type of pollutant). This is a very large and unsubstantiated assumption. The only citation the authors cite to support this claim is an article about worry, not risk—and this article does not assess the role of distance/proximity. This is a very problematic assumption that needs to be described and justified. Some studies do highlight that distance is one factor that affects risk perception, but risk is a much more complex construct than can be proxied by simple distance. There is almost no literature cited/discussed in this manuscript that addresses the complex issue of risk perception.
Line 68: How are distance to a contaminated site and duration of residence “objective exposure measures”? Neither measure necessarily has anything to do with exposure.
Line 71: What do the authors mean by “true perception of environmental exposure”? What does “true” mean if the authors are talking about perception? Do they mean to contrast perceived contamination with empirical or measured contamination? This is confusing.
Lines 73-88. This paragraph addresses the role of children and adolescents in CEC environments. Yet, children and adolescents in the current study make up less than 0.02% of study participants (n=268). I don’t understand the relevance of this paragraph. Perhaps it is explained later on in the paper, but at this point in the manuscript it is unclear.
Lines 222-226. I would be interested to learn more about the FCC distribution. How many cases are in the left tail? If not too many, it likely won’t impact parametric statistics (mean, sd) for comparing the data with those from the national survey—and then one wouldn’t have to use non-parametric tests, which have much less power and cannot be used to generalize beyond the sample (since they are non-parametric). Or simply use the transformed data, described in lines 235-236, for these comparisons. The cubed transformation may take care of the left skew if it’s a worry. If the authors can use parametric statistics, this would allow the authors to include effect sizes for comparisons of difference and support their use of multiple regression (which uses parametric statistics in some computations).
Table 1. We need to see how the FCC study compares to the national survey in terms of demographics so we can get a sense of representativeness and bias. Adding those data and using some chi-square test should help in this table.
Lines 273-282. How do the authors control for the confounding factors listed in this paragraph—such as hours of sleep, alcoholic drinks, smoking, and chronic disease (among others)? There doesn’t appear to be any methods for analysis of confounding factors, such as with the use of part/partial correlations, randomizing study participants into different groups, restricting the sample to exclude certain participants, etc. Maybe this appears later in the paper but it needs to be addressed here or in the methods.
Table 2: If the authors can report the mean and median values in this table, readers will understand the extent and direction of skewness in the distributions.
Table 2: this table reports that “All comparisons were statistically different at p-value ≤0.001.” I have trouble believing this. For example, male vs. female median PCS scores differ by 0.9 with nearly identical interquartile ranges in a sample of n=7873. The authors argue that these are statistically significantly different at the <.001 level. I question this result and many others in the table. Perhaps there is a mistake in the analysis?
Lines 310-311: the authors compare scores to residents living </=2 and >/=3 miles away from the CEC. What about residents who live between 2 and 3 miles (2>x<3)? The selection of groups here seems a bit arbitrary and unusual without any explanation or justification. This creates the primary finding for this study (lines 388-390), so there needs to be some discussion of choices here.
Lines 318-321: the authors report median values and confidence intervals. This is very strange. CI = sample mean +/- z-score. Why are they reporting medians with mean-based CIs? Table 3 suggests this is a mistake and they want to report the interquartile range (IQR). This is confusing.
Lines 328-329 (and elsewhere): the coefficient of determination (R-squared) should be reported as a proportion, not a percentage, e.g., 0.038 instead of 3.8%.
Lines 326-334. The introduction makes it seem like the majority of the analysis in this paper is about the relationship between RP and HRQoL scores. Yet, these results are only reported in 8 lines of text in section 3.4. Out of all the results, “for the FCC adult 327 participants, closest RP had a significant negative association to MCS and PCS.” Nothing else was significant. So in the end, did this study find minimal association between RP and HRQoL? Moreover, no data are reported here, simply p values and R-squared. Where are the slopes/estimates and standard errors for each factor? I think they may be hidden in Supplementary Table B, which is not referenced in the text. Are the reported p values from t tests of slope=0? This is the main part of the study and very little information is included.
Figure 1. The manuscript uses medians and IQR to compare the data in this figure, but the figure uses means and standard error. Why? This is inconsistent.
Lines 388-390: “In this study, we found that living in close proximity to the former uranium processing facility was significantly associated with lower self-reported mental and physical HRQoL scores at the baseline examination for both adult and minors at enrollment participants.” Where are the specific results demonstrating this claim? There is a reference in the text to a supplementary table C, but this appears to have nothing to do with this claim.
Lines 400-403: “Historical research revealed that a large landfill and two major highways outside the 5-mile ring might have negatively impacted the community's self-rated health. Conversely, residents within the 3-mile ring enjoyed proximity to Miami Whitewater Forest park and a golf course, which may have positively influenced their self-rated health.” Why are these factors introduced here at the end of the manuscript? Why were they not considered for development of the hypotheses?
Lines 408-409: “This led us to reclassify the primary independent variable into two groups: residents within 2 miles of the facility and those 3-5 miles away.” Here at the end of the manuscript, readers are told that the analysis they just read about was changed and don’t match the hypotheses laid out at the beginning of the study. This is very confusing. More confusing is that this does not match the text in lines 310-311, which says </=2 and >/=3.
Lines 456-459: the authors state that “the key finding—that environmental contamination increases the risk of diminished mental and physical well-being—is relevant across all communities, including racial or ethnic minorities and socioeconomically disadvantaged groups.” Yet, the study respondents were 99.6% White. Why do they make this spurious claim? Moreover, their “key finding” regards risk of diminished mental and physical well-being. Risk was not the focus of this study. The focus of this study, as stated in the introduction, was about the relationship between residential proximity and HRQoL scores.
Lines 493-494: Here the authors also make conclusions about risk without studying risk: “The findings of this study indicate that residing close to a CEC site poses a risk for diminished HRQoL.”
Author Response
Hello, please see the attached PDF file that contains a table of point-by-point responses to each comment provided by Reviewer 1. We thank this reviewer for their thorough feedback to improve the manuscript.

Reviewer 2 Report
Comments and Suggestions for Authors
While the research is well-structured, there are some areas where improvements are needed to enhance clarity, address potential biases, and strengthen the manuscript’s contribution.
Major comments:
1. Recall Bias and Exposure Measurement:
The reliance on self-reported residential proximity (RP) and duration introduces potential recall bias, particularly given the extended time frame covered by the study.
While the authors mention verifying some addresses with external records, this approach should be emphasized as a limitation. Additionally, the lack of direct environmental exposure data (e.g., air or soil contaminant levels) weakens the robustness of the findings.
2. Study Design and Cross-Sectional Nature:
The cross-sectional design limits causal interpretations. While the authors acknowledge this limitation, it would be valuable to propose specific longitudinal analyses, or follow-up plans to strengthen future research.
3. Statistical Analyses:
The manuscript uses multiple sensitivity analyses to address the relationship between RP, duration, and HRQoL. However, some results lack clarity in presentation. For example, the marginal effects (ME) and regression coefficients for specific subgroups (e.g., minors at enrollment) should be contextualized more thoroughly in terms of practical significance.
Given the heterogeneity in the study population, subgroup analyses by demographic or behavioral factors could provide additional insights.
4. HRQoL Metrics and Comparisons:
The study compares HRQoL scores to the U.S. General Population yet does not provide sufficient detail about the potential differences in demographic or socioeconomic factors between these groups. This comparison may introduce bias if these variables are not adequately controlled.
5. Conceptual Framework:
The discussion should integrate more theoretical perspectives on how environmental contamination impacts mental and physical health. For instance, "place identity" and "environmental stress theories" are briefly mentioned but could be expanded upon to provide a stronger conceptual foundation for the observed results.
6. Vulnerable Populations:
While the study highlights children as a vulnerable group, the analysis and discussion do not adequately explore how these impacts may differ from adults. Further elaboration on how developmental and psychosocial factors contribute to differences in HRQoL would be beneficial.
Minor comments:
Clarity and Readability:
The manuscript’s introduction is comprehensive but could be streamlined to focus on the research gaps and objectives more concisely.
Some technical terms (e.g., "cubed transformation") may need clarification for a broader readership.
Ethics and Data Sharing:
The manuscript states that informed consent was obtained, but further details about how participant confidentiality was maintained would enhance the ethical rigor.
The availability of the Fernald Community Cohort (FCC) data for future research is commendable and should be emphasized as a strength.
Recommendations for Improvement:
· Address recall bias explicitly and consider incorporating objective exposure metrics in future studies.
· Enhance the discussion of theoretical frameworks underpinning the findings.
· Clarify statistical analyses and provide practical interpretations of results.
· Explore subgroup analyses to identify differential impacts on vulnerable populations.
Comments on the Quality of English LanguageGrammar:
Minor grammatical errors and typos were noted (e.g., "its contents are solely the responsibility" should be revised for consistency).
Author Response
Hello, please see the attached PDF file that contains a table of point-by-point responses to each comment provided by Reviewer 2. We thank this reviewer for their thorough feedback to improve the manuscript.

Reviewer 3 Report
Comments and Suggestions for Authors
This is an interesting and well written study.
The results are a little surprising, but the discussion of other factors such as proximity to highway etc do put them in context.
I do have one significant comment which I feel needs to be discussed in a revised manuscript is that prevailing winds etc have not been discussed, so if someone is living 1 mile from the site in one direction, is that the same as living 1 mile in another direction?
Perhaps an additional analysis considering maybe 6 by 6oosegments might be a useful addition, as opposed to just distance.
Even to discuss this aspect would give added value I feel
Author Response
Hello, please see the attached PDF file that contains a table of point-by-point responses to each comment provided by Reviewer 3. We thank this reviewer for their feedback to improve the manuscript.

Round 2
Reviewer 1 Report
Comments and Suggestions for Authors
The revisions to the manuscript are overall excellent and many of the critical typos and missing information have been addressed. I also appreciate the changes to improve clarity. This is somewhat of a complicated study; anything the authors can do to ease readability is appreciated. I have a couple very minor issues that can be considered before publication:
Table 1. In the text, can the authors add a sentence to explain/describe the representativeness of their data to the larger (national) sample?
Table 2. Can the authors note in the text that, while the differences are statistically significant at the .001 level, they are modest. (This would obviate the need to add effect sizes.)
Table 3. Do the authors really mean “less than or equal to .001” or just less than .001?
Supplementary Table B: Some p values are reported to the thousandths place (3 sig figs) and some to the ten thousandths place (4 sig figs). Is there a reason for this difference? If not, perhaps it’s best to select one level of precision and apply it consistently.
Line 321: Is “p<.0001” a typo? Should it be “p<.001”?
Author Response
Hello,
Thank you for your thorough and constructive feedback. We sincerely appreciate your additional comments aimed at enhancing accuracy and readability for a broad audience.
Please find attached a document with our point-by-point responses to each of your critiques.
